# GC-MS Analysis and Biomedical Therapy of Oil from n-Hexane Fraction of *Scutellaria edelbergii Rech*. f.: In Vitro, In Vivo, and In Silico Approach

**DOI:** 10.3390/molecules26247676

**Published:** 2021-12-18

**Authors:** Muddaser Shah, Waheed Murad, Najeeb Ur Rehman, Sidra Mubin, Jamal Nasser Al-Sabahi, Manzoor Ahmad, Muhammad Zahoor, Obaid Ullah, Muhammad Waqas, Saeed Ullah, Zul Kamal, Rafa Almeer, Simona G. Bungau, Ahmed Al-Harrasi

**Affiliations:** 1Department of Botany, Abdul Wali Khan University Mardan, Mardan 23200, Pakistan; muddasershah@awkum.edu.pk; 2Natural and Medical Sciences Research Center, University of Nizwa, P.O. Box 33, Birkat Al Mauz, Nizwa 616, Oman; obaidullah@unizwa.edu.om (O.U.); mwaqas@unizwa.edu.om (M.W.); saeedullah@iccs.edu (S.U.); 3Department of Botany, Hazara University Mansehra, Mansehra 21310, Pakistan; shahhu123@gmail.com; 4Central Instrument Laboratory, College of Agriculture and Marine Sciences, Sultan Qaboos University, Muscat 123, Oman; jamal@squ.edu.om; 5Department of Chemistry, University of Malakand, Chakdara 18800, Pakistan; manzoorhej@yahoo.com; 6Department of Biochemistry, University of Malakand, Chakdara 18800, Pakistan; mohammadzahoorus@yahoo.com; 7Department of Biotechnology and Genetic Engineering, Hazara University Mansehra, Mansehra 21120, Pakistan; 8H.E.J. Research Institute of Chemistry, International Center for Chemical and Biological Science, University of Karachi, Karachi 75270, Pakistan; 9Department of Pharmacy, Shaheed Benazir Bhutto University, Upper Dir 18000, Pakistan; xulkamal@sbbu.edu.pk; 10School of Pharmacy, Shanghai Jiao Tong University, Minhang 800, Shanghai 200240, China; 11Department of Zoology, College of Science, King Saud University, P.O. Box 2455, Riyadh 11451, Saudi Arabia; ralmeer@ksu.edu.sa; 12Department of Pharmacy, Faculty of Medicine and Pharmacy, University of Oradea, 410028 Oradea, Romania; simonabungau@gmail.com

**Keywords:** GC-MS analysis, antibacterial activity, antioxidants, anti-diabetics, anti-inflammatory, analgesic assay

## Abstract

The current study aimed to explore the crude oils obtained from the n-hexane fraction of *Scutellaria edelbergii* and further analyzed, for the first time, for their chemical composition, in vitro antibacterial, antifungal, antioxidant, antidiabetic, and in vivo anti-inflammatory, and analgesic activities. For the phytochemical composition, the oils proceeded to gas chromatography-mass spectrometry (GC-MS) analysis and from the resultant chromatogram, 42 bioactive constituents were identified. Among them, the major components were linoleic acid ethyl ester (19.67%) followed by ethyl oleate (18.45%), linolenic acid methyl ester (11.67%), and palmitic acid ethyl ester (11.01%). Tetrazolium 96-well plate MTT assay and agar-well diffusion methods were used to evaluate the isolated oil for its minimum inhibitory concentrations (MIC), minimum bactericidal concentration (MBC), half-maximal inhibitory concentrations (IC_50_), and zone of inhibitions that could determine the potential antimicrobial efficacy’s. Substantial antibacterial activities were observed against the clinical isolates comprising of three Gram-negative bacteria, viz., *Escherichia coli*, *Klebsiella pneumoniae*, and *Pseudomonas aeruginosa*, and one Gram-positive bacterial strain, *Enterococcus faecalis*. The oils were also effective against *Candida albicans* and *Fusarium oxysporum* when evaluated for their antifungal potential. Moreover, significant antioxidant potential with IC_50_ values of 136.4 and 161.5 µg/mL for extracted oil was evaluated through DPPH (1,1-Diphenyl-2-picryl-hydrazyl) and ABTS assays compared with standard ascorbic acid where the IC50 values were 44.49 and 67.78 µg/mL, respectively, against the tested free radicals. The oils was also potent, inhibiting the α-glucosidase (IC50 5.45 ± 0.42 µg/mL) enzyme compared to the standard. Anti-glucosidase potential was visualized through molecular docking simulations where ten compounds of the oil were found to be the leading inhibitors of the selected enzyme based on interactions, binding energy, and binding affinity. The oil was found to be an effective anti-inflammatory (61%) agent compared with diclofenac sodium (70.92%) via the carrageenan-induced assay. An appreciable (48.28%) analgesic activity in correlation with the standard aspirin was observed through the acetic acid-induced writhing bioassay. The oil from the n-hexane fraction of *S. edelbergii* contained valuable bioactive constituents that can act as in vitro biological and in vivo pharmacological agents. However, further studies are needed to uncover individual responsible compounds of the observed biological potentials which would be helpful in devising novel drugs.

## 1. Introduction

The use of herbs as a source of therapy started since the beginning of human civilizations and till date plays a key role to enhance clinical screening and is also effective in quality control to cope with several health complications [1]. Of the 32,000 documented plants, around 10% have therapeutic significance over the globe [2]. Medicinal plants and their chemical constituents provide the basis of phytomedicines viz antibacterial, antifungal, antioxidant, inflammation, and analgesic potentials [3]. The market demands for herbal medications are regularly increasing and pharmaceutical companies have concerns regarding its global impact and natural therapeutic remedies which can increase the efficacy rate and potency in association with economical, affordable, less toxic, and effectiveness [2].

Plants extracted oils consist of versatile chemical constituents and are well known for their diverse medicinal significance, originating from plants using classical and advanced approaches [4]. Numerous techniques were adopted for the extraction of oils including hydro-distillation, steam-distillation, micro-wave, and organic solvent extraction [5]. Secondary metabolites in the medicinal plants are mostly affected by the plant harvesting period, climatic changes, geographical, and physiological behaviors, and techniques used for essential oils extraction. Regardless of the aforementioned factors, it has also been observed that the composition of the essential and heavy oils varies from one part to another in the same plant [6]. Terpenoids are the major group of essential oils and subcategories as monoterpenes, sesquiterpenes, diterpenes, triterpenes, and tetraterpenes [7]. Essential oils have a key role to resist microbes and also play to scavenge the free radicals, curing inflammation and relieving pain, and thus due to these claimed therapies, modern pharmacists suggest oils as a source of aromatherapy [8].

Nowadays, diabetes’ is the main health concern especially type 2 diabetes mellitus (T2DM) remains a terrible metabolic disorder across the world categorized by hyperglycemia, due to the abnormal regulation of insulin. It is a more complex health problem, which has a very high prevalence rate throughout the world, and, according to the WHO report, around 439 million individuals will be influenced by T2DM in the year 2030 [9]. There are different therapeutic approaches for the treatment of T2DM, in which α-glucosidase inhibitors (AGIs) are the more precise therapeutic approach because they control postprandial hyperglycemia and also provide cardiovascular benefits [10]. The *α*-glucosidase is a hydrolyzing enzyme, mostly present over the brush borderline of the small intestine in humans, and reacts to catalyze the digestion of starch and oligosaccharides into absorbable glucose molecules. Due to which it becomes a quick and effective remedy to cope with T2DM. As clinically approved AGIs temporally recover the blood glucose levels and improve type-2 DM complications along with treatment of obesity, but accomplished with side-effects such as flatulence, diarrhea, and abdominal discomfort [11,12,13,14]. Keeping in mind the side effects of the existing synthetic drugs and especially the crucial role of the α-glucosidase enzyme in hyperglycemia, there is an urgent need to discover safe and effective enzyme inhibitors as an approach to effectively control diabetic disorders [15].

The recent emergence of antibiotic resistance among human pathogens is a major challenge to public health. Similarly, efficacy and related toxicity issues limit the choice of available antibiotics and compel researchers to quest for alternative therapeutic options. Currently, plants are the preferred alternative source to explore bioactive compounds [16]. They are a rich source of phenolics and flavonoids which are an important class of antioxidants and directly inhibit bacterial growth and cause hindrance in their pathogenic activity [17]. Moreover, these oxidant compounds possess anti-inflammatory, analgesic, and anti-diabetic capacities [2].

Plant oils contain valuable chemical constituents that serve as a remedy for inflammations, pain relievers, and other complications that imbalance the normal routine of human beings. Such disorders can be treated with herbal-based therapy having the features to cure inflammation, pain, and regulate blood glucose levels [18]. Furthermore, the most recent advancement in pain remedies, scientists are still in the quest to find out safe and economical approaches to overcome the complication discussed earlier [2].

The genus *Scutellaria* (skull cap) (family: Lamiaceae) consists of around 360 species, distributed, and practiced as the remedial source in traditional medicine in China, Korea, North America, Europe, and Pakistan [19]. More than 300 bioactive compounds are isolated and documented from the genus *Scutellaria* [20], mainly contributed by flavonoids, flavones, glycosides, terpenes, terpenoids, monoterpenes, diterpenoids, and triterpenoids [21] and well known for their diverse biological applications against antibacterial, antifungal, antioxidant, antidiabetic, anti-inflammatory and analgesic significance [21,22]. Some species of *Scutellaria* have also been used to cure cancer, hypertension, allergy, sedative, insecticidal, diabetes, and obesity [19]. *Scutellaria edelbergii Rech*. F. locally called panra and practiced for curing inflammation and pain. The fresh leaves are used in making green tea (kawa), and as a blood purifier, widely spread in the temperate and tropical mountains tails such as Europe, North America, and East Asia, and Pakistan. In Pakistan, this species is distributed over the tail of mountains in Swat, Kalam, and Chitral [2], which prefer to grow in habitats with moist loamy soil. The available literature lacks the chemical composition of the oil isolated from the n-hexane fraction of *S. edelbergii*; hence, the current study was focused to explore the chemical composition and highlight its role as aromatherapy by investigating it against the in vitro biological and in vivo pharmacological profiling to add scientific validity to the literature regarding the oil of *S. edelbergii.*

## 2. Results and Discussion

Generally, oils are complex mixtures of essential secondary metabolites isolated from plants, animals, and microbes. There are approximately 10–60 constituents at different concentrations in these mixtures, but usually, 2–4 major compounds attributed the biological properties [23].To see the importance of the oil obtained from the n-hexane fraction of *S. edelbergii*, the current research work was designed to determine the chemical constituents and evaluate its biological, and pharmacological potentials. The study will update, and validate the aroma therapeutic importance, and add a further implication to the biopotency. Plant species from the same genus *Scutellaria* have been previously investigated for the oil composition and their therapeutic importance where it has been concluded that the *Scutellaria* genus plays a key role in curing various ailments, however, this still requires further scientific authentication [4].

### 2.1. GC-MS Analysis

The crude oils extracted from the n-hexane fraction of *S. edelbergii* was profiled via GC-MS. The oils contained 42 bioactive constituents. The higher concentration of linoleic acid ethyl ester (19.67%) was found in the oil followed by ethyl oleate (18.45%), linolenic acid, methyl ester (11.67%), palmitic acid, ethyl ester (11.01%), and hexadecanoic acid methyl ester (7.16%) (Table 1 and Figure 1). The other minor constituents detected in the oil were 13-docosenoic acid, methyl ester (Z) (3.98%), octadecanoic acid, ethyl ester (2.33%), Methyl stearate (1.41%), 2,4-di-tert-butylphenol (1.29%), docosanoic acid, ethyl ester (1.19%), and 1-octadecene (1.01%). The oil has a higher concentration of linoleic acid ethyl ester, previously reported in the essential oils from nine *Platycodi Radix* populations by He et al. [24], and the hydro alcoholic extract of *Cynodon dactylon* reported by Shabi et al. [25] whereas, the flowers of *Coreopsis tinctoria* by Kim et al. [26], while the aerial parts of *Helleborus bocconei subsp. intermedius* described by Rossellia et al. [27]. Ethyl oleate (synonym: 9-octadecenoic acid, ethyl ester) was also determined previously in the essential oils of nine *Platycodi Radix* populations (45.31–10.87%) in the literature described by He et al. [24], green microalga *Chlamydomonas reinhardtii* documented by Herrera-Valencia et al. [28], *Catharanthus roseus* by Lawal et al. [29], the aerial parts of *Helleborus bocconei subsp. intermedius* Rossellia et al. [27], and flowers of *Coreopsis tinctoria* screened by Kim et al. [26]. Similarly, linolenic acid, methyl ester documented by the Lawal et al. [29] and Rossellia et al. [27], palmitic acid, ethyl ester by Udourioh et al. [30] and Cakir et al. [31], and hexadecanoic acid methyl ester [24,27,28] were previously detected in the essential oils of different medicinal plants.

Ethyl linoleate (synonym: linoleic acid ethyl ester) is an unsaturated fatty acid used in many cosmetics for its various attributes, such as antibacterial, anti-inflammatory presented by Park et al. [32], and wound healing determined by Jelenko et al. [33], and clinically proven to be an effective anti-acne agent by Charakida et al. [34]). Recently, it was reported that ethyl linoleate (synonym: linoleic acid ethyl ester) inhibits melanogenesis through the Akt/GSK3β/β-catenin signal pathway mentioned by Ko et al. [35]. Therefore, we suggest that the oil, having a higher concentration of ethyl linoleate, ethyl oleate, linolenic acid, methyl ester, palmitic acid, and ethyl ester, may be useful as a safe whitening agent in cosmetics and a potential therapeutic agent for reducing skin hyperpigmentation in clinics. Moreover, other bioactive compounds as thymol, dodecanoic acid, ethyl ester, and phytol acetate are also present, which are well known for their medicinal values [36]. The aforementioned chemical constituents are already reported from *S. brevibracteata* [37], and *S. heterophylla* [38], belonging to the genus *Scutellaria*. These bioactive compounds have numerous therapeutic significances such as antimicrobial, antioxidant, analgesic, and anti-inflammatory [39].

Compounds identified through elution order on the HP-5MS column; while the RI was identified from the database (NIST, 2011).

### 2.2. Antibacterial Activity

The oils obtained from *S. edelbergii* were tested against human pathogenic bacterial strains (*K. pneumonia*, *P. aeruginosa*, *E. coli*, and *E. faecalis)* and were evaluated for MIC, MBS (shown in Table 1), optical density (OD 600 nm), and half-maximal inhibitory concentrations (IC_50_) using various concentrations (Figure 2A,B), respectively. However, the IC_50_ values were shown for 1/2MIC, MIC, and 2MIC concentrations against various bacteria. The zone of inhibition was also determined at low to high doses (50 µg/mL and 100 µg/mL) represented as dose 1 and 2 and compared with a standard (Levofloxacin) and negative control (DMSO) as shown in Figure 3A. At MIC, the IC_50_ values of *S. edelbergii* against *K. pneumonia* were 81.82%, *P. aeruginosa* 85.06%, *E. coli* 85.20%, and *E. faecalis* 81.58%. The HEOs exhibited an appreciable activity against all the tested bacterial strains. However, among the investigated bacterial strains, the maximum zone of inhibition (14.9 ± 0.01 and 19.2 ± 0.05 mm) at doses 1 and 2, respectively, were displayed against the *E. coli*, followed by (14.7 ± 0.03 * and 17.8 ± 0.03 * mm) *E. faecalis*. The present data showed that *E. coli* is more susceptible to the essential oils of *S. edelbergii*. The considerable activity of the HEOs against the bacteria might probably be due to thymol, 2,4-Di-tert-butyl phenol, hexadecanoic acid, and methyl ester as described by Escobar et al. [40]. Pentadecanoic acid and ethyl ester have been described as antibacterial by Wei et al. [41], whereas methyl stearate has been described by Elshafie et al. [42]. Our data also agree with the reported data of Yu et al. [43], where the oils of *S. barbata* were tested against different bacterial strains. Gram-positive strains are more susceptible to oil compared to Gram-negative bacterial strains, which agrees with the findings of Kasaian et al. [4].

### 2.3. Antifungal Assay

The oils of *S. edelbergii* produced an appreciable effect against the tested fungal strains; *C. albican* and *F. oxysporium* in a dose-dependent fashion. The MIC and MBC of the respective fungal strains are shown in Table 2, where its OD 600 nm and IC_50_ are shown in Figure 2C,D, respectively. The IC_50_ of *S. edelbergii* at MIC was 94.00% and 81.89%, respectively, for *C. albicans* and *F. oxysporum.* Similarly, the maximum zone of inhibition (Figure 3B) was observed against *C. albiacn;* 13.9 ± 0.11 * and 18.7 ± 0.11 * followed by *F. oxysporium;* 13.2 ± 0.22 * and 17.6 ± 0.33 * mm for dose 1 and 2, respectively, in comparison to the standard clotrimazole; 17.3 ± 0.33 and 22.8 ± 0.66 mm at the same concentrations. The antifungal effect is due to the existence of the bioactive compounds present in the oil such as 2,4-di-tert-butyl phenol, thymol, myristic acid, ethyl ester, hexadecanoic acid, methyl ester, and many other chemical compounds have been described by Escobar et al. [40], whereas, for thymol and myristic acid ethyl ester, it has been documented by Huang et al. [44]. Our findings acceded with the data revealed by Yu et al. [43] and Zhu et al. [45] in the literature available for *S. barbata* and *S. strigillosa*, respectively. Our result consented to the data documented by Zahra et al. [36] which screened *S. multicaulis* and *S. bornmuelleri* for the mentioned studies. The similarity was due to the same assay and fungal strains used that were reported for species belonging to the same genus *Scutellaria* and *F. oxysporium* in using the dose-dependent method in comparison with the standard and negative control (Figure 3B). Resistance against the fungal strains due to the existence of the bioactive compounds present in the oil such as 2,4-di-tert-butyl phenol, thymol, myristic acid, ethyl ester, hexadecanoic acid, methyl ester, and many other chemical compounds is described by Escobar et al. [40] for thymol and documented for myristic acid ethyl ester according to Huang et al. [44]. Our findings acceded with the literature data revealed by Yu et al. [43] and Zhu et al. [45] available for *S. barbata* and *S. strigillosa*, respectively. Our result consented to the data documented by Zahra et al. [36] which screened *S. multicaulis* and *S. bornmuelleri* for the mentioned studies. The similarity was due to the same assay and fungal strains used reported for the species belonging to the same genus *Scutellaria.*

### 2.4. Antioxidant Assay

The antioxidant capacity of the oils at concentrations of (1000, 500, and 250 µg/mL) was examined against the in vitro antioxidant capacities using DPPH and ABTS bioassays. The essential oils presented significant antioxidant potential in the DPPH assay with an IC_50_ of 136.4 in comparison with an IC_50_ of 161.5 µg/mL in the ABTS assay; while the antioxidant standard ascorbic acid (AA) displayed IC_50_ values of 44.49 and 67.78 µg/mL for the DPPH and ABTS assays, correspondingly (Figure 4). The free radical scavenger potential was due to the presence of thymol, 2,4-Di-tertbutyl-phenol (Zhao et al. [46]), dodecanoic acid, ethyl ester [47], Linoleic acid, methyl ester, linolenic acid, and methyl ester (Luís et al. [48]). Our finding for antioxidant potential was similar to some *Scutellaria* species: *S. immaculata*, *S. ramosissima*, and *S. schachristanica* [37]. Moreover, our listed data were also in favor of Zahra et al. [36] which reported the essential oils of *S. multicaulis* and *S. bornmuelleri* for their antioxidant activity.

### 2.5. In Vitro Antidiabetic Assay

The oils from *S. edelbergii* displayed potent inhibitory potential with an IC_50_ of 5.45 ± 0.42 µg/mL by screening them in vitro for antidiabetic effect while comparing with standard acarbose (IC_50_ = 377.00 ± 1.06 µg/mL) (Figure 5A,B, respectively). The oils have a significant role in regulating the blood glucose level which was previously described by Yani et al. [49]. Moreover, Tahir et al. [50] also found that oils are a good therapeutic source and act as antidiabetic agents. Yen et al. [51] also validated the in vitro antidiabetic effect of the oils produced by medicinal plants. The constituents present in the mentioned screening are also present in *S. edelbergii* such as thymol which acts as an antidiabetic agent described by Escobar et al. [40] and 2-methylhexacosane reported by Ali et al. [52]. The data documented from our results agree with the literature available for *S. orientalis* and *S. articulata* listed by Formisano et al. [53]—since the selected species belong to the same genus. Meanwhile, our results contradicted those of Jelassi et al. [54] which is stated in the in vitro antidiabetic activity against *Acacia millissima* and *Acacia cyclops* due to the variation in the plant family. Heghes et al. [55] also revealed the importance of oils as a therapy to cure metabolic disorders required for normal physiological activities of humans. To further elucidate all 42 compounds’ interaction with the alpha-glucosidase active site, molecular modeling and docking were performed.

### 2.6. α-Glucosidase Structure Modeling

The crystal structure of *Saccharomyces cerevisiae*, α-glucosidase, has not yet been reported; therefore, a homology model was constructed from the template (PDB ID 3A47) with 72% identity and 99% query coverage. Five models were constructed via MODELLER in which model 3 showed the lowest DOPE score of 72,413.56250, as reported in Table 3. Model 3 was selected as the final model for further validation (Figure 6A–C). The active site and catalytic residues of 3A47 and alpha-glucosidase are conserved (ASP214, GLU276, and ASP349) for substrate attachment (Figure 6C). The homology model of alpha-glucosidase was aligned and superimposed with the template (PDB ID 3A47 [56], having an RMSD value of 1.233 Å as shown in Figure 7.

### 2.7. α-Glucosidase Model Validation

It is essential to validate the homology model of the protein to know about the overall structure of the model. The Ramachandran plot shown in Figure 8 of the model depicts good stereochemical properties. Moreover, 94.197% of the residues of the model are in the highly favorable region, while 5.222% are in the allowed region and 0.580% are in the disallowed area, which are two residues (ALA278 and THR566) away from the active site of the protein. The overall structure quality overview showed that the model is within the acceptable quality zone (Figure 6). The local overall quality of the model residues is reported in Figure 9A. The slight fluctuations are in a positive energy state, which shows the unstable regions of the protein while the overall model is in the negative energy window, showing an overall good quality of the protein model. Figure 9B shows different NMR and X-ray structures compared with the current model of the alpha-glucosidase. The current model shows that the X-ray structures zone has the same amino acid size, showing excellent quality of the model. The high energy and lower energy regions are reported in Figure 9C, where the most inferior energy regions are noted in blue while the red color represents the high energy states. The overall model was registered in the lower energy state, determining the stability of the protein structure.

### 2.8. Docking Simulations

Based on the GC-MS results, potential compounds (42) were identified for molecular docking and docking was performed using the active pocket of the target protein. Based on protein–ligand interactions, the potential compounds were chosen for further validations. Among which 18 compounds presented interactions with the functional residues of the protein active pocket, from which 8 compounds with positive docking scores were excluded. Moreover, the top 10 compounds were screened as final inhibitors for the alpha-glucosidase assay. The final compounds represent interactions with the critical active pocket residues of α-glucosidase (Figure 10). The essential residues ASP214 and ASP349 are involved in the binding interaction with the selected compounds in the vicinity of the active binding site. HIS348 shows a high population rate while interacting with six out of ten compounds in the rocking run (Figure 11), while ASP214 is in the 2nd position in the interaction population—interacting with five compounds. Figure 12 reflects the interactive atoms of each selected compound having a reference substrate alpha-D-glucose with the active site residues of the alpha-glucosidase protein. A different color scheme was used to present residues of the active binding site of the target protein. The 3D interaction of each compound is reported with every 2D image of all selected compounds in Figure 12. The substrate interacts with the active site residues having a refine pose of 0.793Å, while the selected compounds also interact with the important residues of the protein active pocket.

### 2.9. Binding Energy and Binding Affinity

The final 10 selected compounds’ interactions and binding energies reflect the potential inhibitors for the α-glucosidase protein (Table 4). The substrate of the alpha-glucosidase protein is α-D-glucose having a docking score of −3.66 kcal/mol, while the binding energy calculated was −33.49 kcal/mol and binding affinity of −6.35 kcal/mol. The docking score comp-1 shows the highest docking score (−6.27 kcal/mol), a binding energy of −39.87 kcal/mol, and a binding affinity of −8.33 kcal/mol. In the case of compound 2, it has the highest binding energy (−50.55 kcal/mol) with a docking score of −5.28 kcal/mol and the-second highest binding affinity (−9.74 kcal/mol). The second leading target is comp-10, with a binding energy of −47.79 kcal/mol and the highest binding affinity of −9.81 kcal/mol. Comp-8 possesses the third-highest binding energy (−43.75 kcal/mol) with a binding affinity of −8.76 kcal/mol. Comp-3 and comp-4 have a binding energy of −41 kcal/mol with a change in the binding affinity of −8.88 kcal/mol and −9.10 kcal/mol. In the case of comp-6 and comp-7, both compounds have a binding energy of −40 kcal/mol slightly different from that of comp-3 and comp-4, while comp-6 has a binding affinity of −9.2 kcal/mol and comp-7 has a binding affinity of −8.68 kcal/mol. Comp-5 has a binding energy of −38 kcal/mol; the lowest binding energy is shown by comp-9 (−25.49 kcal/mol), and the lowest binding affinity −6.39 kcal/mol. These binding interactions of selected compounds—with the functional residue of the active site of the target protein, binding energies calculated at the best-bounded conformation, and binding affinities—reflect that these compounds can be leading inhibitors for the α-glucosidase protein.

### 2.10. Anti-Inflammatory Activity

The anti-inflammatory capacity of the crude oils from *S. edelbergii* is listed in Table 5. Doses at a concentration of 5, 10, and 15 mg/kg were used which produced an appreciable inhibition from low to high dose 48.22, 58.15, and 61% inhibition, respectively, in comparison to the standard diclofenac sodium which possessed (70.92%) inhibition against the decrease in the paw diameter in the tested experimental animals caused by carrageenan, whereas the control normal saline did not affect paw edema. Santos et al. [57] also described that the oils are accountable to inhibit paw edema. The constituent required to cope with the inflammation is thymol, which is present in the understudied plant and was reported previously by Escobar et al. for its pharmacological effect [40]. This significant activity was due to the presence of the responsible compounds such as hexadecanoic acid and methyl ester, which cure inflammation, as documented by Kumar et al. [58], and linoleic acid ethyl ester, phytol, and acetate, by Huang et al. [44]. Our current results regarding the essential oils of *S. edelbergii* agree with the reported results of Zahra et al. [36] for some species of the same genus.

### 2.11. AnalgesicActivity

The analgesic capacity of the oils of *S. edelbergii* was determined in the experimental animals which are given in Table 6. Low to high dose was used at (5, 10, and 15 mg/kg) presented a significant 26.61, 41.82, and 48.28% inhibition; while aspirin, which was used as a standard aspirin, produced a 59.69% reduction in writhes in comparison with acetic acid which caused writhes in the mice. However, the normal control demonstrated no activity. The main compound among the essential oils which has the property of relieving pain was octadecanoic acid, ethyl ester in the literature of Whittle et al. [59], ethyl oleate, and Heneicosane described by Asgarpanah et al. [60], and perhaps some others which require screening. The literature reported by Quintao et al. [61] shows that the essential oils contain some valuable bio-constituents which can reduce pain. Hajhashemi et al. [62] also reported that the oils have the capacity to cure the pain due to various bioactive constituents such as 2,4-Di-tert-butylphenol described by Zhao et al. [46]; whereas, the chemical constituents ethyl-oleate, hexadecanoic acid, butyl ester, octadecanoic acid, ethyl ester, and heneicosane, previously described by Akin-Osanaiye et al. [63], Kumar et al. [64], and Asgarpanah et al. [60], are also the constituents of the *S. edelbergii* which are responsible for curing pain. Thus, the mentioned studies of the oil of *S. edelbergii* agree with the literature reported for genus *Scutellaria* by Yin et al. [65] which are also similar to *S. lateriflora* presented by Uritu et al. [66].

## 3. Material and Methods

### 3.1. General Instrumentation

Glass column for column chromatography (CC) via silica gel 60 (70–230 mesh ASTM, Merck KGaA, Darmstadt, Germany); analytical grade methanol, n-hexane, and dimethyl sulfoxide (DMSO 99%) (Fisher Scientific, Loughborough, UK). Buchi Rotavapor R-210 (Shimomeguro, Tokyo, Japan); DPPH (2,2-diphenyl-1-picrylhydrazyl), ABTS, (2,2-azino-bis(3-ethylbenzothiazoline-6-sulfonic acid), carrageenan, diclofenac sodium, nutrient agar and potato dextrose agar (Sigma-Aldrich Chemie GmbH, Taufkirchen, Germany); gas chromatography-mass spectrometer (GC-MS-QP2010, Shimadzu, Kyoto, Japan).

### 3.2. Plant Material

The plant specimens of *S. edelbergii* were collected during the flowering season at (April–June 2019) from the mountain’s tails (1660–2200 m) of Kalam, District Swat, Khyber Pakhtunkhwa, (KPK) Pakistan, and identified by the taxonomist Prof. Mehboob Rahman Matta College Swat (MCS), Khyber Pakhtunkhwa, Pakistan. The preserved sample was deposited at the Herbarium Department of Botany, Abdul Wali Khan University, Mardan with a voucher specimen No. (AWKUM/Herb/2234) for future studies. The raw materials of *S. edelbergii* were washed, dried under the shade, and powdered using an electric blender. The obtained (2 kg) plant powder was then kept in the refrigerator at 4 °C [2].

### 3.3. Soaking, Extraction, and Fractionation

The selected plant powder was then soaked with (4 L) analytical grade methanol (MeOH), in glass containers for three weeks followed by continuous shaking. The solubilized plant material was then passed through a muslin cloth into a flask and proceeded to a rotary evaporator at ±40 °C for crude extract preparation. The paste obtained via Rotavapor was air dried, eventually yielding 600 g of crude extract and transferred into an airtight glass bottle. For fractionation of the extract, 500 g of the crude extract was submerged into 0.4 L distilled water in a 1000 mL glass beaker and homogenized via stirring, which was then transferred to a separating funnel for solvent-solvent extraction according to the polarity gradient (low to high polar) using solvents n-hexane, chloroform (CHCl₃), ethyl-acetate (EtOAc), and n-butanol (n-BuOH). The solvents that remained in each fraction were evaporated under maintained temperature and rotation per minute of the rotavapor at 40 °C and 80 rpm correspondingly. Each fraction n-hexane, CHCl₃, EtOAc, n-BuOH, and the aqueous yields 21, 19, 20, 18, and 35 g dry mass, respectively.

### 3.4. Oil Extraction and Preparation of FAEs

The obtained n-hexane fraction was further subjected to column chromatography and eluted with n-hexane solvent as a mobile phase (Figure 13) to obtain a crude oil. The column was continuously run with pure n-Hexane to obtain maximum oil. Total fatty acids in crude oils were transformed into their corresponding esters, through esterification as described by Zhang et al. [67] and He et al. [24]. Two milliliters of 5% H_2_SO_4_/CH_3_OH was added to a 3 mL diluted solution of *S. edelbergii* crude oil, reacted at 70 °C in a water bath for 30 min, then extracted twice with 1.5 mL of n-hexane using a vortex mixer for 30 s. The superior phase contained total FAEs, which was further proceeded for GC-MS analysis to observe the complete profile.

### 3.5. GC-MS Analysis

The crude oils obtained from the n-hexane fraction of *S. edelbergii* was profiled using gas chromatography-mass spectrometry (GC-MS) analysis, coupled with a Perkin-Elmer Clarus 600 GC Scheme, which was fitted with Rtx-5MS, a very small capillary column (30 m × 0.25 mm I.D × 0.25 μm with film thickness; up to 260 °C), fixed to a Perkin-Elmer Clarus 600 MS. Ultra-high purity of helium (99.99%) was administrated as a carrier gas at a constant maintained flow rate of 1.0 mL/min. The injection temperature was 260 °C, while the temperature transfer line was 270 °C. The ion source was observed with a temperature of 280 °C. Furthermore, the ionizing energy (IE) was 70 eV. Moreover, the electron multiplier (EM) voltage was accessed from auto-tune. All data were obtained by assembling the full-scan mass spectra within the scan range of 45–550 a.m.u. The sample (at a concentration of 1 μL) was injected with a split ratio of 10:1. Moreover, the oven temperature was fixed for 1 min at 60°C and a specific rate of temperature of 4°C/min up to 260°C was maintained and fixed for 4 min. The system took 50 min to complete its total run [23].

### 3.6. Detection and Identification of Compounds

The volatile chemical compounds of *S. edelbergii* (whole plant) were profiled using GC-MS analysis and the unknown natural products were authenticated via MS library software (NIST (2011 version 2.3): National Institute of Standards and Technology, Mass Spectra Libraries, Gaithersburg, MD, USA, and Wiley MS, 9th ed.) [68]. Moreover, the use of RI (obs.) and its comparison with RI (lit.) was also used for the identification of the chemical constituents. Further confirmation of aliphatic hydrocarbons was carried out by referring to RI data generated from a series of n-alkanes (C9–C40) [23]. The quantification was performed using an external standard technique, whereas the calibration curves were produced by operating the GC analysis of the representative bioactive compounds [23].

### 3.7. Stock Solution Preparation

The oils extracted from *S. edelbergii* were dissolved in DMSO at 1 mg/5 mL and 2 mg/5 mL represented as doses 1 and 2, respectively, for biological activities using the previously reported method [5].

### 3.8. Antimicrobial Screening

The in vitro antimicrobial assays were analyzed using *K*. *pneuomoniae*, *P. aeruginosa*, *E. coli*, and *E. faecalis*, and human pathogenic clinical isolates strains and fungal *C. albican* and *F. oxysporum* strains were taken from the Microbiology laboratory following safety roles, authenticated by Chairperson, Dr. Hazir Rahman, Department of Microbiology, AWKUM, Mardan. MIC was determined through the 96-well plate method by measuring the OD600 nm, where its MBC was calculated through the agar plate method, where the zone of inhibitions (ZOIs) of the HEO was performed using the agar well diffusion method.

#### 3.8.1. MIC/MBC Assay

The respective strains (*K. pneuomoniae*, *P. aeruginosa**, E. coli*, and *E. faecalis*) were sub-cultured on a fresh nutrient agar plate for 24 h before antibacterial assays. The inoculum was prepared by transferring a single colony of microbes to a sterile nutrient broth media, kept overnight at 37 °C on a shaker bed at 220 rpm. The OD600 nm of the bacterial suspension was measured by UV–Vis spectroscopy (Cary100Bio)—which was adjusted to 0.5 McFarland standards. MIC of the HEOs of *S. edelbergii* was determined by using the tetrazolium microplate assay for concentrations 1, 5, 10, 15, 25, 50, 100, 150, 200, and 500 µg/mL (2-fold serial dilution in sterile broth). Then, 100 µg/mL of each concentration was incubated in triplicate wells along with 20 µL of respective bacterial inoculum (1.5 × 10^6^ CFU) for 18–24 h at 37 °C ± 0.5. After incubation, in each well, 50 µg/mL of 3-(4,5-dimethylthiazol-2-yl)-2, 5-diphenyltetrazolium bromide (0.2 mg/mL) MTT was added, and the plate was incubated at 37 °C for 45 min. An appropriate solvent blank (sterile broth) was included as a negative control and the bacterial suspension was included as the positive control. The absorbance was measured at 570 nm and a reference wavelength of 650 nm by adding sterile nutrient broth on a spectrophotometer and the percentage reduction in the dye (indicating the bacterial growth inhibition) was calculated by IC_50_ = OD of the positive control-OD of the test sample/OD of the positive control × 100. Similarly, minimum bactericidal concentrations (MBC) for all these tested samples were determined by the agar plate method, in which 100 µL from MIC results of all concentrations were coated on TSA plates and cultured overnight. Zero growth showed MBC.

#### 3.8.2. Zone of Inhibitions

The ZOIs were determined at dose-dependent concentrations represented as doses 1 and 2 which were taken from the already prepared stock solution and resolubilized in DMSO using a standard approach [9]. The required materials (nutrient agar media, steel wire loop, borer, and glass Petri plates) for the antibacterial assay were autoclaved at 121 °C for 20 min, then under hygienic conditions in the laminar flow hood, each Petri dishes was filled with 20 mL nutrient agar media and left undisturbed to solidify. The available (*K. pneumoniae*, *P. aeruginosa*, *E. coli*, and *E. faecalis*) bacterial strains were properly inoculated at a concentration of 1.5108CFU/mL of the bacterial cell density (BCD), over the solidified media using a wire loop as per accordance with the Microbiology Lab AWKUM rules. In the Petri dishes, four wells—at equal distances of 3 mm—were made. The essential oils at dose 1 (concentration of 50 µg/mL) were injected using a micropipette into the 1st and 2nd wells, while the negative control (DMSO) and Levofloxacin standard were employed to the 3rd and 4th wells, respectively. The same procedure was carried out for dose 2 (100 µg/mL). The Petri dishes were properly covered and kept in the incubator at ±37 °C for 24 h and then the zone of inhibition was recorded in mm. All data were taken in triplicate and were statistically analyzed and represented as the mean ± SEM.

### 3.9. Antifungalassay

Antifungal activity on the crude oils of *S. edelbergii* was determined by evaluating MIC/MBC and ZOIs by using the tetrazolium 96-well microplate method, agar plate, and agar well diffusion assays by using the procedure mentioned in Section 3.8.1 and Section 3.8.2 (Methodology section). The respective fungal strains (*C. albicans and F. oxysporum*) were sub-cultured on fresh potato dextrose agar (PDA) plates for 24 h before antifungal assays. MIC/MBC were determined for the oil of *S. edelbergii* at concentrations of 1, 5, 10, 15, 25, 50, 100, 150, 200, and 500 µg/mL, where, for the ZOIs, dose 1 (50 µg/mL) and 2 (100 µg/mL) were taken from stock solutions and resolubilized in DMSO. This assay was conducted using potato dextrose agar (PDA) media and sterile broth using the reported method [2]. The PDA (39 g) was mixed with 1 L distilled water, shaken till completely homogenized, covered, and placed along with the required apparatus (media, Petri dishes, steel borer, and loop), and autoclaved at 121 °C for 20 min. For ZOIs determination, each of the glass Petri dishes were filled with 20 mL nutrient media under a laminar flow hood and then allowed to solidify. The fungal inoculum at a concentration of 10^8^–10^9^ CFU/mL was added over the solidified media and four holes using the standard method were made for the injection of doses 1 and 2, negative control (DMSO), and standard clotrimazole, accordingly. Then, the plates were incubated for 72 h at 25 °C in the incubator. Next, the zone of inhibition was measured in mm. For the scientific validity and authenticity, all data were taken in triplicates and listed as the mean ± SEM.

### 3.10. Antioxidant Activity

#### 3.10.1. DPPH Assay

The significance of the crude oils to scavenge the free radicals was determined via DPPH and ABTS assay using the reported method [69]. To proceed with the DPPH assay, 3000 µg of the DDPH was homogenized in 100 mL distilled methanol (DM) and then kept in the dark for 30 min so that the solution could produce free radicals. Various dilutions of the essential oils and standard ascorbic acid (1000, 500, and 250 µg/mL) were prepared. Then, 2 mL each (HEOs and ascorbic acid) was added with 2 mL of the prepared DPPH solution and was kept incubated in the dark for 25 min. After incubation, the absorbance of the essential oils and standard ascorbic acid was computed at 517 nm via UV/Vis spectrophotometry. Equation (1) was used to analyze the antioxidant significance of the EOs and standard.
(1)% Scavenging activity=A−B÷A×100
where *A* = Control absorbance; *B* = Standard absorbance (AA).

#### 3.10.2. ABTS Assay

To determine the antioxidant potential of the HEOs, an ABTS assay was also carried out using a previously reported approach. To proceed with the assay, ABTS at a quantity of 383 mg and 66.2 mg potassium persulfate (K_2_S_2_O_8_) were each separately homogenized at 100 mL analytical grade MeOH and then mixed. Then, 2 mL from the reagent mixture (ABTS) was incubated with 2 mL of the essential oils and standard (AA) for 25 min in the same concentrations as mentioned in the DPPH assay. The absorbance of the sample (HEO and AA) was measured at 746 nm via UV/Vis spectrophotometry. The antioxidant significance was calculated using Equation (1).

### 3.11. Antidiabetic Activity (α-Glucosidase Assay)

The crude oils were also examined against the in vitro α-glucosidase significance using (50 mM) phosphate buffer pH (6.8). The enzyme was properly dissolved in the phosphate buffer; 1 U/2 mL, 20 µL/well of the enzyme, and 135 µL/well phosphate buffer was used as reaction buffer, 20 µL/well of the oils were solubilized in DMSO (0.5 µg/mL), in 96-wells plates incubated for 15 min at 37 °C. After the incubation, the substrate *p*-nitrophenyl-α-D-glucopyranoside at a concentration of 0.7 mM was added and the change in the absorbance was estimated for 30 min at 400 nm. The 7%, DMSO, and acarbose were used as a positive control and standard, respectively [70]. To predict the leading inhibitors among the selected small compounds from the crude, molecular docking and binding energy calculation approach were used.

### 3.12. 3 D Structure of α-Glucosidase

The GC-MS-detected compounds of *Scutellaria edelbergii* interaction with the α-glucosidase of *Saccharomyces cerevisiae* was conducted. The homology model of the α-glucosidase protein was assembled. The protein sequence was retrieved from the GenBank with server given as (https://www.ncbi.nlm.nih.gov/genbank, accessed on 27 October 2021) with accession number “CAA85264” [71]. The protein sequence was blasted against the Protein Data Bank (PDB) server (https://www.rcsb.org, accessed on 27 October 2021) for template identification via NCBI protein blast (blastp). The template selected for the homology model (PDB ID 3A47 [56]) showed 99% query coverage and 72% sequence identity. MODELLER [72] software was used to construct the homology model of the protein. The sequence and query were aligned, and five models were built. The final model was selected on the lowest DOPE score of the MODELLER. The homology model was validated through the Ramachandran plot server (https://zlab.umassmed.edu/bu/rama, accessed on 27 October 2021) [73] for the protein’s stereochemical properties ProSA-web server for the overall quality score of the model compared with the crystal structures of the proteins. Hydrogen atoms were added to the protein model, and partial charges were calculated using the tethered energy minimization of molecular mechanics via the molecular operating environment (MOE-2020.0109) autocorrect protocol to fix the bad contacts and imperfect geometries. To determine the local minima of the protein (low energy conformations), energy minimization was performed using the AMBER14: ETH force field.

### 3.13. Docking Simulations

To predict the preferred orientation of two chemical species interactions, the molecular docking simulation technique was used. The MOE s dock application was used to predict favorable binding modes of the small molecule with the macromolecular target (protein). Each small molecule placement in the active pocket, known as poses, was created and scored. The scoring function of the poses is the binding free energy of the small molecule with the protein in terms of entropy and solvation energies, polar interaction energies enthalpy, or qualitative shaped-based numerical assessment. The poses with the highest scoring and conformational energies were used to perform further analysis.

To perform the molecular docking simulations of the 42 compounds isolated through GC-MS analysis, the MOE was used. The fast Fourier transform (FFT) protocol was selected to generate thousands of poses of a small molecule around the protein’s active site. The London dG solvation model’s R-Field electrostatics was chosen for the refined top 30 poses using Equation (2).
(2)$$\Delta G=c+E_{flex}+\underset{h-bonds}{\sum }c_{HB}f_{HB}+\underset{m-lig}{\sum }c_{M}f_{M}+\underset{atom\;i}{\sum }\Delta D_{i}$$

In Equation (2) “*c*” calculates the gain or loss in the entropy (rotational/transitional), loss of energy in the ligand is calculated by E_flex_, the imperfections in the hydrogen bonds are calculated by f_HB_. In contrast, the favorable hydrogen bond energy is calculated by cHB, the metal ligands imperfection is calculated by fM. In contrast, the ideal metal-binding energy is calculated by c_M_, and Di calculates each atom dissolution energy.

The top 30 poses retrieved from the London dG algorithm were refined to select the final pose via GBVI/WSA dG solvation method using Equation (3), which is implemented in MOE’s Dock.
(3)$$\Delta G\approx c+a\left[\frac{2}{3}\left(\Delta E_{coul}+\Delta E_{sol}\right)+\Delta E_{vdw}+\beta \Delta SA_{weighted}\right]$$

In Equation (3), the increase or decrease in the entropy (rotational/transitional) is “*c*”, “α and β” are the constants dependent on the force field selected, coulombic electrostatic are calculated by E_Coul,_ the solvation energies are calculated E_sol_, van der Waals contributions of the systems are E_vdW,_ and SA weighted calculates the exposed surface area.

The structures of 42 compounds identified from the *S. edelbergii* essential oils were retrieved from the PubChem [74] database using the names of the compounds, and each compound 3D structure was downloaded in SDF format and saved in the MOE database. The cleaning of each molecule in a database is desirable before performing any analysis. The database wash algorithm implemented in MOE applies a set of rules to clean each molecule, which contains extraneous salts—removing or adjusting states of protonation to ensure each structure is at appropriate shape for the protein–ligand docking.

Forty-two compounds selected for molecular docking were washed and energy minimized via the MMFF94X force field designed for small molecules to gain the minimal energy state of each compound. The docking protocols reported above were applied to the molecular docking of the 42 selected compounds with the active pocket of the α-glucosidase. The final refine pose for each small compound was saved in the MDB format with the energies calculated in the database. A total of 18 compounds showed interaction with the alpha-glucosidase active site residues. The top ten compounds were selected as potent inhibitors for the α-glucosidase active pocket, based on the molecular docking score generated from the GBVI/WSA dG algorithm and the best interactions with the key residues of alpha-glucosidase functional pocket α-glucosidase were further selected for binding energy and binding affinity calculation.

### 3.14. Binding Energy and Binding Affinity Calculation

To identify the binding potential of a small molecule with the protein or the energy required to detach the small molecule from the active pocket of the protein, binding energy and binding affinity were calculated. The minimization of the active pocket was performed each time, calculating the binding energy and binding affinity. The small drug-like inhibitors were kept flexible to move freely with the active site residues. The binding energy and binding affinity were calculated using the MOE’s implicit solvent method generalized Born/volume integral (GB/VI). Generalized Born interactions are non-bonded interaction energies among proteins and small-molecule inhibitor-containing coulomb electrostatic interactions. All the energies were calculated in kcal/mol.

The finally identified potential ten compounds were categorized based on an excellent inhibitory effect with alpha-glucosidase based on the docking score (binding energy) and binding affinity. They can be further subjected to the experimental validation of each compound.

### 3.15. In Vivo Pharmacological Activities

This current study is part of a continuous project on Biomedical applications of *S. edelbergii*, and is the first to determine the significance of HEOs of the selected plant against in vivo anti-inflammatory and analgesic bioassay using Swiss albino mice as experimental tools, sanctioned by the authorized ethical committee end rosed No: AWKUM/Bot/2019/1720, on dated: 29 January 2019) reported previously by Shah et al. [2] and given in hardcopy to the publisher.

#### Experimental Animals

According to ethics approval (No: AWKUM/Bot/2019/1720, dated: 29 January 2019), healthy Swiss albino mice (24–30 g) were purchased from the Veterinary Research Institute, Peshawar (VRI), and were housed in rubber cages at the AWKUM animal house under hygienic conditions for 45 days and maintained at a temperature of ±20 °C. The ARRIVE guideline was followed to fed animals (rodent pellets, food, and water) as well as the inclusion of doses, standard, control, and samples size using standard protocol [75].

### 3.16. Anti-Inflammatory Activities

To determine the anti-inflammatory activity of the oils of *S. edelbergii,* thirty (30) Swiss albino mice were taken as experimental animals and equally distributed into six groups; each group contained (*n* = 5) Swiss albino mice:

Carrageenan (1 mL) as an inducer of paw edema was infused into each group (1st–6th). Then, after 30 min, 1 mL of normal saline (NS) was injected into the mice of the second group to evaluate their effectiveness, and then 1 mL diclofenac sodium (DS) as a standard was infused to compare the mice of group three. However, after 30 min of inducer injection, doses of the HEO at a concentration of 5, 10, and 15 mg/kg/bodyweight were inoculated to the Swiss albino mice of the fourth, fifth^,^ and sixth groups, respectively.

To evaluate the significance of the tested samples, experimental animals’ paw diameter was measured and noted after 1, 2, and 3 h, respectively. The anti-inflammatory % inhibition was calculated as per Equation (4).
(4)Percent inhibition=A−BA×100
where A) inducer (carrageenan); B = inhibition of the tested sample (crude oils, standard, and control); whereas in analgesic activity A) represents the writhes inducer acetic acid.

### 3.17. Analgesic Activities

The analgesic potential of the crude oils was determined via an acetic acid-induced writhing assay in Swiss albino mice. Initially, thirty (30) experimental animals were taken and equally distributed into six groups each having five (5) Swiss albino mice. The dose, control, and standards were all administrated to the experimental animals by mean of intraperitoneal muscle using an approved size sterilized syringe. One milliliter of acetic acid was injected into each group (1st, 2nd, 3rd, 4th, 5th^,^ and 6th) and then after 30 min, 1 mL of normal saline (as a normal control) was injected into the mice of the second group, followed by the injection of 1 mL aspirin as a standard to the third group and HEO at a concentration of 5, 10, and 15 mg/kg B.W was infused to the 4th, 5th and 6th Swiss albino mice groups, respectively.

The number of writhes was counted in comparison with the standard drug and control for 10 min to estimate the capacity of *S. edelbergii* HEO and the obtained result was statistically analyzed and represented as % inhibition, following Equation (4).

### 3.18. Statistical Analysis

All the data of the current project were taken in triplicate and calculated via one-way analysis of variance (ANOVA), through Bonferroni’s test (at significance level: *p* = 0.05 and 0.01) and two-way ANOVA, through Sidak’s multiple comparisons test (*p* = (ns > 0.9999, **** < 0.0001)) to obtain statistically valid results. Whereas, for antioxidant potential, a nonlinear regression graph (NLRG) was plotted among % inhibition and concentration of the tested drug, and the IC_50_ was calculated using GraphPad Prism 9 software for Windows (GraphPad-Software, San Diego, CA, USA, 2020) using the following equation:*Y* = 100/1 + (*ˆHill Slope*)

where 1 = represents inhibitors concentration; Y = indicates the inhibitor’s reaction; Hill Slope shows the steepness of the curves.

## 4. Conclusions

Plant-extracted oils are gaining traction in the pharmaceutical industry as well as in the food, cosmetics, and perfumery industries. Their wide use as antioxidant, antimicrobial, antifungal, anxiety agents, and pain relievers makes them an alternative and useful herbal therapeutic remedy. The oil of *S. edelbergii* offered considerable antibacterial and antifungal activity against human pathogenic bacterial and fungal strains in comparison to the standard. These HEO bioactive metabolites may have multi-targets on antimicrobial cells, which may provide an alternative approach for antimicrobial resistance and associated future threats and crises. The oil also depicted an appreciable free radical scavenger effect using DPPH and ABTS free radicals. The significant in vitro glucosidase activity also suggests the oil of *S. edelbergii* as intoxicating to cure diabetes. The HEO was also found effective against the in vivo anti-inflammatory and analgesic bioassays. Furthermore, based on our systematic analysis, the oil of *S. edelbergii* could be employed as an antimicrobial, antioxidant, antidiabetic, anti-inflammatory, and analgesic agent, due to the presence of biologically active constituents. However, additional investigations are necessary to highlight and isolate more bioactive compounds accountable for examining in vitro and in vivo biological activities.

## Figures and Tables

**Figure 1 molecules-26-07676-f001:**
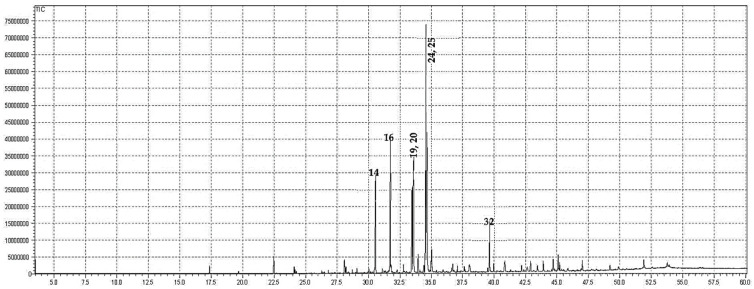
GC chromatogram of the oils of *S. edelbergii*.

**Figure 2 molecules-26-07676-f002:**
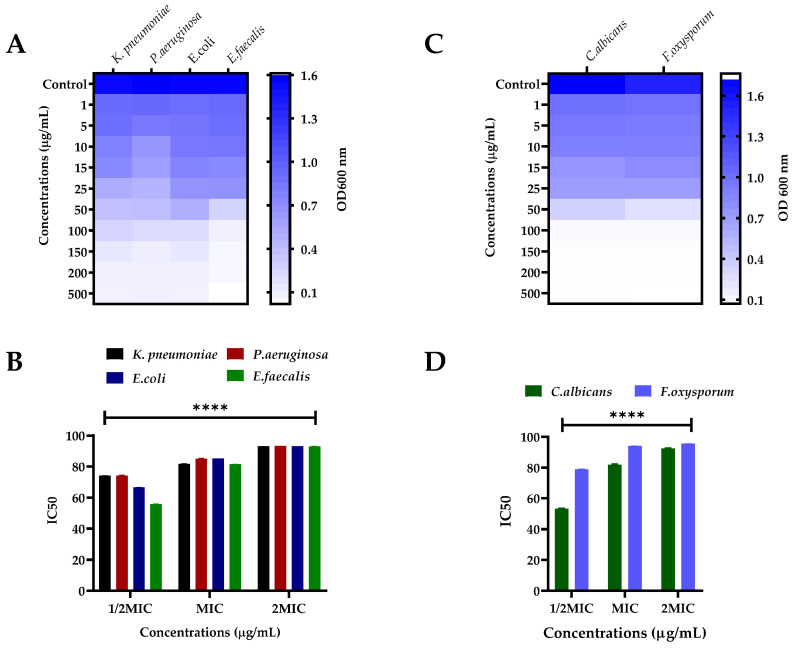
(**A**) Antibacterial OD 600 nm and IC_50_ of *Klebsiella pneumoniae*, *Pseudomonas aeruginosa*, *Escherichia coli*, *Enterococcus faecalis* against various concentrations of crude oils of *S. edelbergii* and (**B**) MIC of the bacterial strains selected at various concentrations *(***C**) antifungal OD 600 nm and IC_50_ of the potential of oils of *S. edelbergii* against *Candida albicans* and *Fusarium oxysporum* against various concentrations Whereas, as (**D**). Represent the MIC against the fungal strains used. All Data were taken in triplicate (*n* = 3) and analyzed through two-way ANOVA, via Tukey’s multiple comparison test, ns = > 0.9999, **** *p* = < 0.0001).

**Figure 3 molecules-26-07676-f003:**
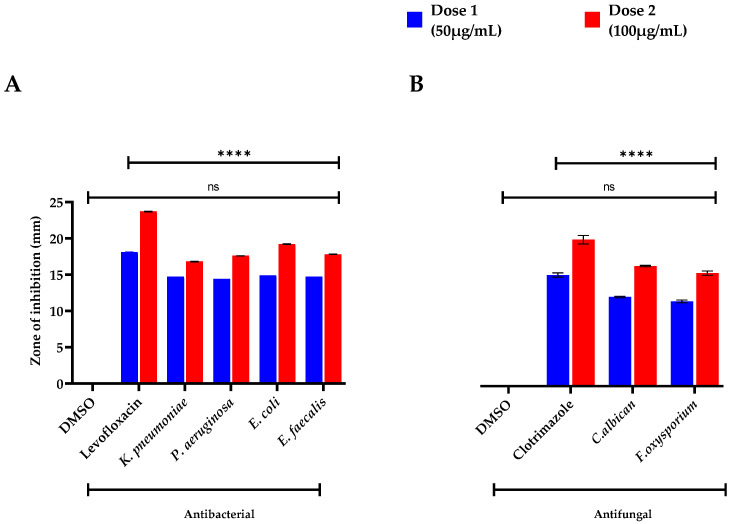
(**A**) Antibacterial and (**B**) antifungal zone of inhibitions of oil of *S. edelbergii*. DMSO = negative control, Levofloxacin and Clotrimazole = positive control. Data were taken in triplicate (*n* = 3) and analyzed through two-way ANOVA, via Sidak’s multiple comparison test, ns = > 0.9999, **** *p* = <0.0001).

**Figure 4 molecules-26-07676-f004:**
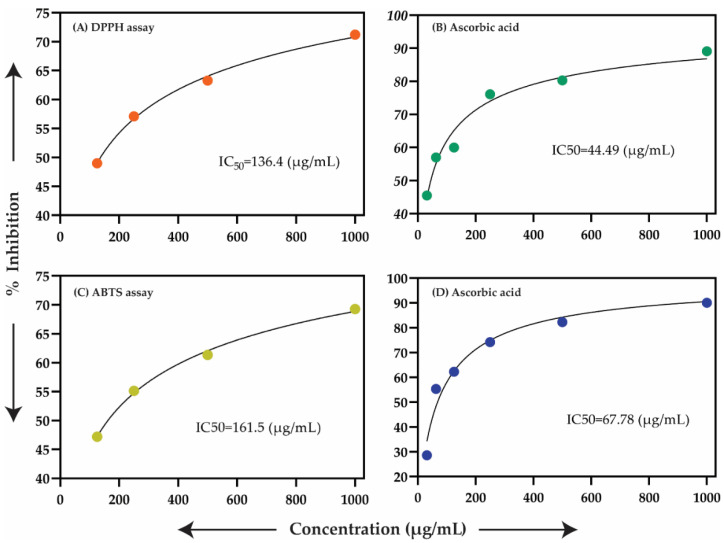
Antioxidant activity of oil of *S. edelbergii* where (**A**) represents the antioxidant activity of the oils of *S. edelbergii*; (**B**) the potential of ascorbic acid (standard) using DPPH assay; (**C**) displays the antioxidant activity of the oils using ABTS assay; (**D**) the potential of ascorbic acid via ABTS assay.

**Figure 5 molecules-26-07676-f005:**
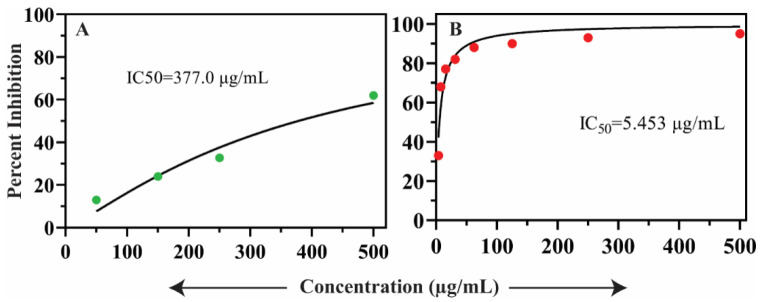
% inhibition of α-glucosidase activity (**A**) oil of *S. edelbergii* and (**B**) acarbose (standard).

**Figure 6 molecules-26-07676-f006:**
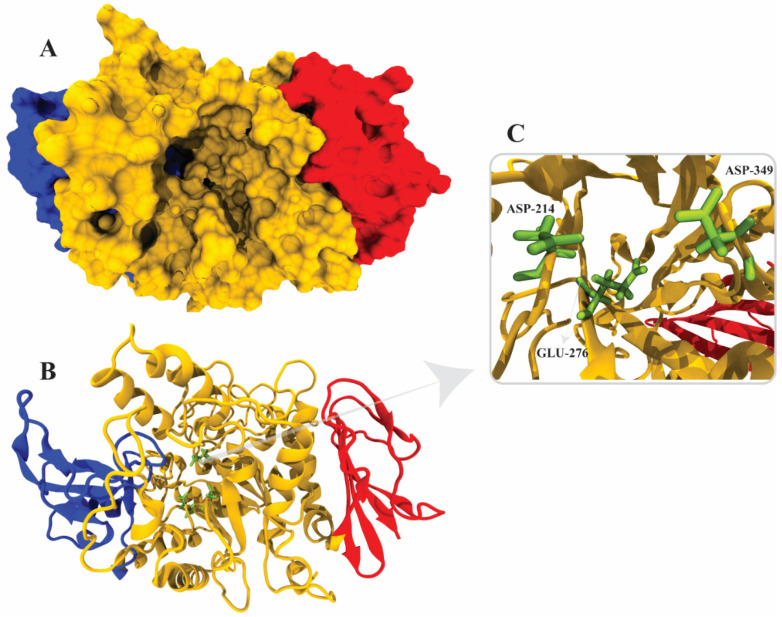
Three-dimensional structure of α-glucosidase, (**A**) surface representation, (**B**) cartoon representation defining the catalytic resides in green color, (**C**) active site residues of α-glucosidase. The yellow color represents domain A (amino acids 1–113 and 190–512), the blue color represents domain B (amino acids 114–189), and the red color represents domain C (amino acids 513–589).

**Figure 7 molecules-26-07676-f007:**
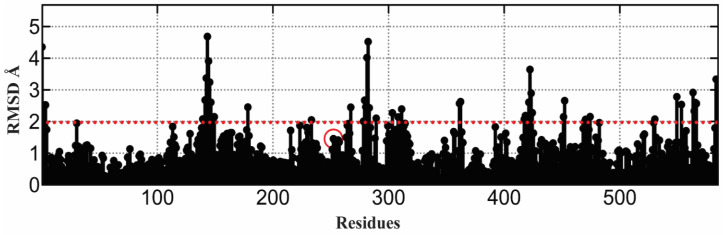
Homology model of α-glucosidase aligned and superimposed with template RMSD in angstroms. The red line shows the template RMSD values while the red circles identified the catalytic residues.

**Figure 8 molecules-26-07676-f008:**
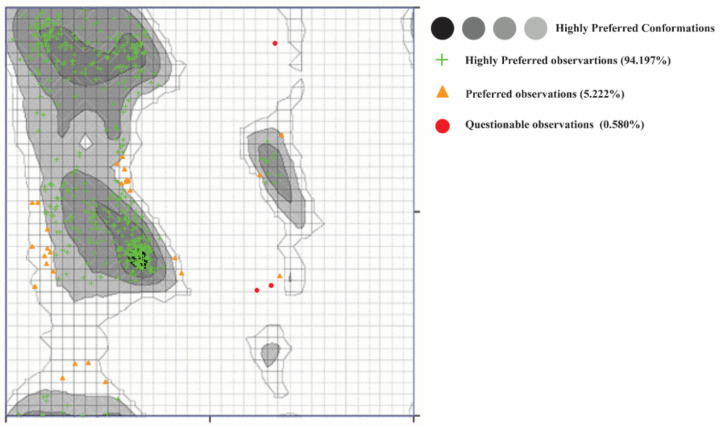
Ramachandran plot of α-glucosidase represents the highly favorable regions in green, allowed region in orange, and disallowed residues in red.

**Figure 9 molecules-26-07676-f009:**
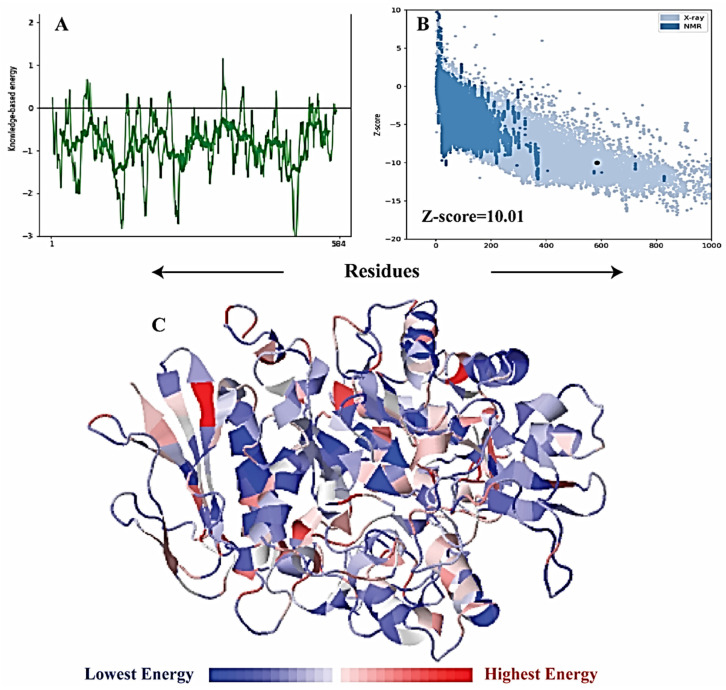
Overall quality of the α-glucosidase model from the ProSa-Web server. (**A**) Model residue; (**B**) NMR and X-ray structures comparison with the current model of the alpha-glucosidase; (**C**) high energy and lower energy regions.

**Figure 10 molecules-26-07676-f010:**
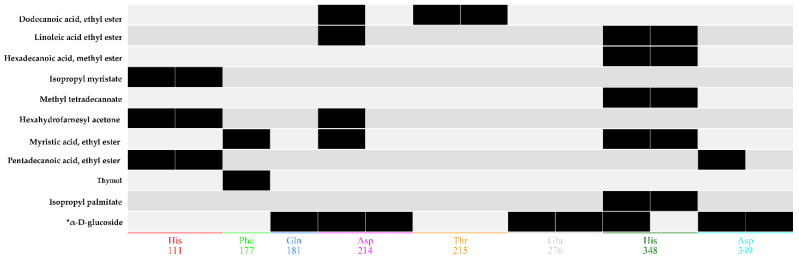
Barcode representation of the final 10 compounds with the reference substrate α-D-glucose, interacting with the alpha-glucosidase active site residues. Each black barcode represents the hydrogen bond interaction of the compounds in *y*-axis with the active side residue in the *x*-axis. * Reference substrate (α-D-glucose) of the alpha-glucosidase protein interaction with active site residues.

**Figure 11 molecules-26-07676-f011:**
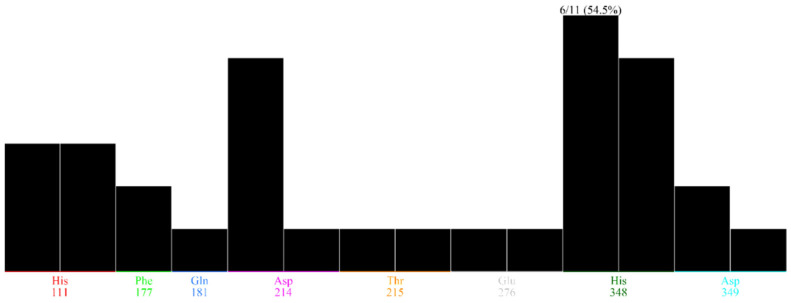
Overall percentage of hydrogen bond interactions of each active pocket residue with the selected inhibitors. His348 interact with 50% (6 compounds out of 10) of the selected compounds, while Asp214 form interactions with the 40% (5 compounds out of 10).

**Figure 12 molecules-26-07676-f012:**
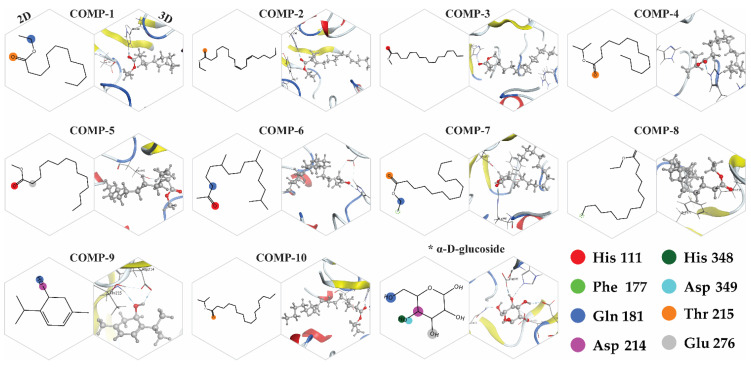
Two-dimensional and three-dimensional interaction representation of the selected 10 inhibitors having a reference substrate (alpha-D-glucose) with the active pocket residues of the α-glucosidase protein. The interactive atoms of each compound with a specific residue are colored as the residue assigned color in the 2D representation. * The alpha-glucosidase protein substrate (α-D-glucose) interaction with active site residues.

**Figure 13 molecules-26-07676-f013:**
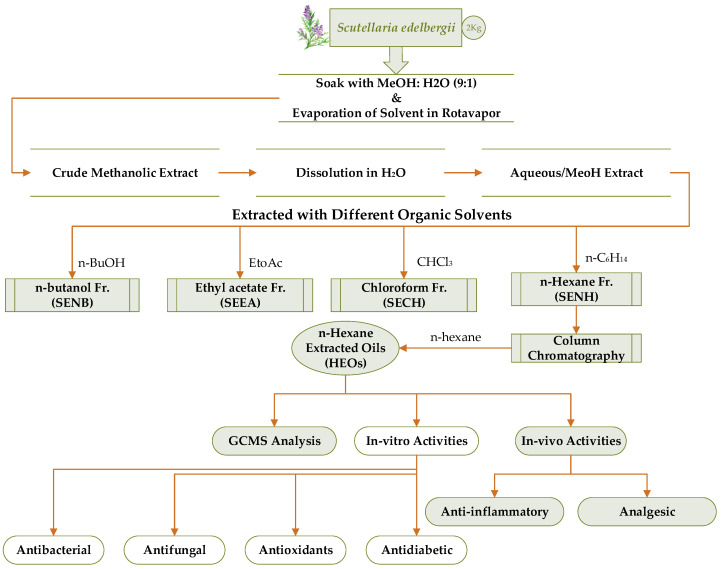
Schematic representation of the current study (soaking, extraction, fractionation identification, in vitro, and in vivo activities of oil). SENH = *S. edelbergii* n-hexane, SECH *= S. edelbergii* chloroform, SEEA = *S. edelbergii* ethyl acetate, SENB = *S. edelbergii* n-butanol, SEAQ = *S. edelbergii* aqueous fraction.

**Table 1 molecules-26-07676-t001:** Identification of compounds by GC-MS analysis in the oils of *S. edelbergii*.

Compounds	Contents (%)	RI_calc._	RI_rep._
Thymol	0.61	1295	1266
1-Tridecene	0.18	1307	1287
2,4-Di-tert-butylphenol	1.29	1518	1519
Cetene	0.47	1593	1587
Dodecanoic acid, ethyl ester	0.29	1596	1581
Methyl tetradecanoate	0.23	1727	1714
1-Octadecene	1.01	1793	1788
Myristic acid, ethyl ester	0.47	1796	1778
Octadecane	0.52	1800	1800
Isopropyl myristate	0.26	1824	1827
Hexahydrofarnesyl acetone	0.34	1845	1842
Pentadecanoic acid, ethyl ester	0.11	1896	1878
Methyl hexadec-9-enoate	0.09	1908	1892
Hexadecanoic acid, methyl ester	7.16	1929	1908
Ethyl 9-hexadecenoate	0.126	1976	1955
Palmitic acid, ethyl ester	11.01	1997	1968
Isopropyl palmitate	0.13	2026	2011
Heptadecanoic acid, methyl ester	0.18	2029	2008
Linoleic acid, methyl ester	7.02	2099	2071
Linolenic acid, methyl ester	11.67	2107	2077
Oleic acid, methyl ester	0.38	2111	2082
Methyl stearate	1.41	2130	2133
Dodecyl nonyl ether	0.66	2158	2158
Linoleic acid ethyl ester	19.67	2168	2177
Ethyl oleate	18.45	2173	2171
Hexadecanoic acid, butyl ester	0.15	2189	2174
Octadecanoic acid, ethyl ester	2.33	2197	2181
Heneicosane	0.41	2100	2100
Phytol, acetate	0.13	2123	2222
cis-11-Eicosenoic acid, methyl ester	0.64	2306	2284
Eicosanoic acid, methyl ester	0.48	2331	2310
13-Docosenoic acid, methyl ester, (Z)	3.98	2509	2473
Docosanoic acid, methyl ester	0.71	2533	2502
Docosanoic acid, ethyl ester	1.19	2598	2576
Tricosanoic acid, methyl ester	0.12	2634	2612
2-Methylhexacosane	0.61	2700	2664
15-Tetracosenoic acid, methyl ester, (Z)	0.23	2712	2709
Tetracosanoic acid, methyl ester	0.38	2735	2712
(E)-3,7-Dimethylocta-2,6-dien-1-yl palmitate	0.75	2756	2747
Squalene	0.82	2837	2808
1,6,10,14,18,22-Tetracosahexaen-3-ol, 2,6,10,15,19,23-hexamethyl-, (all-E)-	0.14	2890	3059
n-Nonacosane	0.98	2900	2900
Identified compounds	97.79		

RI_calc._ = retention index calculated, RI_rep._ = retention index reported.

**Table 2 molecules-26-07676-t002:** MIC and MBS of the crude oils of *S. edelbergii* against various bacterial and fungal strains.

Strains		MIC (µg/mL)	MBC (µg/mL)
Bacteria	*K. pneumoniae*	100	150
	*P. aeruginosa*	100	150
	*E. coli*	100	150
	*E. faecalis*	50	100
Fungi	*C. albicans*	100	100
	*F. oxysporum*	50	100

**Table 3 molecules-26-07676-t003:** DOPE profiling of homology models created by MODELLER.

Model	MolPDF	DOPE Score
Model-1	3649.48315	72,730.43750
Model-2	3525.77832	73,327.53906
Model-3	3936.56714	72,413.56250
Model-4	3989.72241	72,646.79688
Model-5	3680.53467	72,542.36719

**Table 4 molecules-26-07676-t004:** Binding energy and binding affinity were calculated of 10 selected compounds having a reference substrate molecule (α-D-glucose) with the α-glucosidase protein active site.

S. No	Name	CID	Energies = Kcal/mol		
			Docking Score	Binding Energy	Binding Affinity
1	Dodecanoic acid, ethyl ester	7800	−6.27	−39.87	−8.33
2	Linoleic acid ethyl ester	5,282,184	−5.89	−50.55	−9.74
3	Hexadecanoic acid, methyl ester	8181	−5.28	−41.29	−8.88
4	Isopropyl myristate	8042	−4.82	−41.10	−9.1
5	Methyl tetradecanoate	31,284	−4.73	−38.65	−8.64
6	Hexahydrofarnesyl acetone	10,408	−4.45	−40.55	−9.2
7	Myristic acid, ethyl ester	31,283	−4.22	−40.40	−8.68
8	Pentadecanoic acid, ethyl ester	38,762	−4.01	−43.75	−8.76
9	Thymol	6989	−3.61	−25.49	−6.39
10	Isopropyl palmitate	8907	−3.19	−47.79	−9.81
11	α-D-glucose	7,9025	−3.66	−33.49	−6.35

**Table 5 molecules-26-07676-t005:** Anti-inflammatory activity of crude oils of *S. edelbergii*.

Changes in Paw Diameter (Mean ± SEM)						
Treatment	Conc. (mg/kg)	After 1 h	After 2 h	After 3 h	Aver. Measurement	% Inhibition
Inducer (Carrag)	1 mL	1.15 ± 0.01	1.39 ± 0.03	1.71 ± 0.03	1.41 ± 0.03	
NS	1 mL	1.14 ± 0.03	1.39 ± 0.03	1.70 ± 0.03	1.41 ± 0.03	-
DS	10	0.54 ± 0.05	0.41 ± 0.03	0.29 ± 0.5	0.41 ± 0.02	70.92
HEO	5	0.79 ± 0.07	0.73 ± 0.01	0.67 ± 0.03	0.73 ± 0.05 *	48.22
	10	0.71 ± 0.04	0.59 ± 0.02	0.47 ± 0.06	0.59 ± 0.04 *	58.15
	15	0.68 ± 0.06	0.56 ± 0.03	0.42 ± 0.01	0.55 ± 0.03 *	61

Carrag = carrageenan, NS = normal saline, diclofenac sodium = positive control, HEO = n-hexane extracted oil, *n* = 3 with *p* ≤ 0.05 *; data were taken as the mean ± SEM.

**Table 6 molecules-26-07676-t006:** Analgesic significance of *S. edelbergii* crude oils.

Treatment	Dosage (mg/kg)	Counted Writhes Mean ± SEM	% Reduction (Writhes)
AA	1 mL	26.3 ± 0.05	
NS	1 mL	26.1 ± 0.05	-
Aspirin	1 mL	10.6 ± 0.03	59.69
HEO	5	19.3 ± 0.04 **	26.61
	10	15.3 ± 0.05 **	41.82
	15	13.6 ± 0.02 **	48.28

AA = acetic acid, NS = normal saline, HEO = n-hexane extracted oil, standard = aspirin; data were taken in triplicates and represented as the mean ± SEM at significance (*p* ≤ 0.01 **).

## Data Availability

The data is available in the Appendix A to the researchers.

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
