# Peer review of "GC-MS Analysis and Biomedical Therapy of Oil from n-Hexane Fraction of Scutellaria edelbergii Rech. f.: In Vitro, In Vivo, and In Silico Approach"

_molecules, 2021, doi:10.3390/molecules26247676_

Round 1
Reviewer 1 Report
In this paper authors studied n hexane extracted oil fraction of scutellaria and further analyzed pharmacological properties of fractions.
All studies were correctly performed.
pls check English in introduction
Hence this reviewer indicate accept with minor revision this paper for publications in pharmaceuticals
Author Response
Reviewer 1
Comments and Suggestions for Authors
In this paper, the authors studied the n-hexane extracted oil fraction of Scutellaria and further analyzed the pharmacological properties of fractions.
All studies were correctly performed.
Reviewer Comment: pls check the English in the introduction
Author response: The introduction was re-checked and removed typographical mistakes.
Hence this reviewer indicates accept with minor revision this paper for publications in pharmaceuticals.
Author response: Worthy reviewer thanks for your appreciation
Reviewer 2 Report
I don't know if concerning so called haexan oil You repet column chromatography?It is totaly different from previous decribed method.
I want to belive that is was repetead.
Author Response
Reviewer 2nd
Comments and Suggestions for Authors
Reviewer Comment: I don't know if concerning so-called hexane oil You repeat column chromatography? It is different from the previously described method.
I want to believe that is was repeated.
Author response: Worthy reviewer, although the obtained mixture contained fatty acids and fatty acid esters but our target was to characterize and identify volatile components. Worthy reviewer, you better know that GC-MS can only detect volatile components which in this sense have given the name of essential oil. To further clarify these are not an essential oil, in that sense where it is obtained by hydro-distillation. This oil (mostly a mixture of fatty acids and fatty acid esters) is obtained after passing the hexane fraction through column chromatography. After packing and loading the column with the slurry of n-hexane fraction, we passed n-hexane solvent to remove fatty acid and fatty acid esters and then increased the solvent system polarity for the isolation of diterpenoids, triterpenoids, and other constituents. During column chromatography, we got a good quantity of oil with hexane solvent, and the only way to see the chemical constituents is GC-MS analysis. That is why we decided to do the GC-MS after showing promising activity. On your worthy suggestion we have changed the title as well.
The reference cited in the manuscript does not belong to us. It is a typographical mistake. We have revised the paragraph in the revised version.
Reviewer 3 Report
This manuscript is the research about GC-MS analysis and biomedical therapy of oil obtain Scutellaria edelbergii Rech. f.: in vitro, in vivo, and in silico approach, which is interesting research, but there are some points to be clarified
Line 263-274, This study uses mixtures as the research object, although there are GC-identified ingredients as supporting evidence, the relationship between the test results and the dose has not been converted in accordance with the proportion or established an estimation standard. Therefore, the use of literature review to replace specific experimental data has almost no basic scientific rigor, and it is difficult to verify the academic value of the manuscript, which should improve the result and discussion.
Line 277-321. The search for α-glucosidase and 2-methylhexacosane related research is only limited to ref-33 (https:// doi.org/10.1016/j.sjbs.2021.04.060). Based on the previous review comments, this manuscript has an "Overexplain" phenomenon in the scientific connection between the Hexane Extraction Oil (HEO) and the discussion literature.
Line 377-388; Line 391-402. As suggested by the above review, the author is requested to provide a specific literature basis, especially the composition and dosage, as the basis for discussion of this manuscript.
Author Response
Reviewer 3rd
Comments and Suggestions for Authors
Reviewer Comment: This manuscript is the research about GC-MS analysis and biomedical therapy of oil obtain Scutellaria edelbergii Rech. f.: in vitro, in vivo, and in silico approach, which is interesting research, but there are some points to be clarified.
Author response: Thanks for your kind suggestions. The corrections have been made in the revised version.
Reviewer Comment: Line 263-274, This study uses mixtures as the research object, although there are GC-identified ingredients as supporting evidence, the relationship between the test results and the dose has not been converted in accordance with the proportion or established an estimation standard. Therefore, the use of literature review to replace specific experimental data has almost no basic scientific rigor, and it is difficult to verify the academic value of the manuscript, which should improve the result and discussion.
Author response: Worthy reviewer, research on phytochemicals is carried out in a well-organized manner, first the crude extract is screened for various in vitro biological potential, further they characterized through HPLC, GS-MS and LC-MS etc the prediction of the known compounds. Then the crude extract is fractionated to make them easy for isolation. Worthy reviewer, as you know plants contain million of compounds and in a single lab or researcher cannot isolate all of them to screen them individually for different biological potentials therefore, the preliminary results about the fraction and crude extract are published to attract the attention of scientific community for focusing their research on the specific plant which make the research easy and in comparatively short time. In this broader sense in our current research is on Scutellaria edelbergii, we have got the crude oil which was screened for in-vitro activities, after showing promising results the oils were further subjected for various in-vivo pharmacological activities. We are working now on its ingredients to be isolated and will do all these studies on those compounds. So, these are our preliminary studies for anti-inflammatory, antibacterial, and anti-fungal potentials of Scutellaria species. This plant is investigated by very few researchers and hopefully from this publication we would be able to attract the scientific community to do research on this plant. The established research is carried out on isolated compounds. Hopefully we will get them soon.
Reviewer Comment: Line 277-321. The search for α-glucosidase and 2-methylhexacosane related research is only limited to ref-33 (https:// doi.org/10.1016/j.sjbs.2021.04.060). Based on the previous review comments, this manuscript has an "Overexplain" phenomenon in the scientific connection between the Hexane Extraction Oil (HEO) and the discussion literature.
Author response: worthy reviewer, as per your suggestion more literature has been added for variations in the revised version.
Reviewer Comment: Line 377-388; Line 391-402. As suggested by the above review, the author is requested to provide a specific literature basis, especially the composition and dosage, as the basis for discussion of this manuscript.
Author response: Valuable Reviewer thanks for your suggestion. Literature has been added in the revised version for further validation.
Round 2
Reviewer 3 Report
the authors have answered my comments
This manuscript is a resubmission of an earlier submission. The following is a list of the peer review reports and author responses from that submission.
Round 1
Reviewer 1 Report
This paper describe study of activity of hexan extracts of Scutellaria and analysis of composition
All the significant studies such MTT cytotoxicity, MIC and MBC analysis, IC50 determination were made. Also GC-MS were performed to demonstrate phytochemical composition.
Docking simulations for glucosidase were performed but need to improve it. Pls also add positive reference in table 4.
Fig 10 and 11 are not much clear.....suggest to modify format.
Correct abstract font size (first part is different that final part). Punctuation and space have to be checked with attention before to submit paper….(i.e. fig 3 lack: A)!!!
Hence, this reviewer indicate accept this MS for publication after minor revision in Molecules.  
Author Response
Response to Reviewer 1st
Comments and Suggestions for Authors
Reviewer Comment: This paper describes the study of the activity of hexane extracts of Scutellaria and analysis of the composition
All the significant studies such as MTT cytotoxicity, MIC, and MBC analysis, IC50 determination were made. Also, GC-MS was performed to demonstrate phytochemical composition.
Author response: Worthy reviewer thanks for appreciating our work.
Reviewer Comment: Docking simulations for glucosidase were performed but need to improve. Pls also add the positive reference in table 4.
Author response: Dear reviewer thanks for your valuable suggestion it will further improve our article. The standard has been added in the table 4. The comments have been addressed and highlighted in the revised manuscript.
Reviewer Comment: Fig. 10 and 11 are not much clear.....suggest modifying the format.
Author response: Dear reviewer we have focused your valuable comments and addressed accordingly in the revised manuscript.
Reviewer Comment: Correct abstract font size (the first part is different from that final part). Punctuation and space must be checked with attention before submitting paper….(i.e., fig 3 lack: A)!!!
Author response: Worthy reviewer, yes the font size was not the same as per your suggestion corrected and in (figure 3) A was missed which is inserted in the revised manuscript.
Hence, this reviewer indicates accepts this MS for publication after minor revision in Molecules.
Author response: Dear reviewer, All the comments have been addressed accordingly and highlighted in the revised manuscript and thanks for considering our work for publication in this scientific journal.

Reviewer 2 Report
The process of extracting the plant material and the process of obtaining the hexane fraction (so called by the authors as oil) is unclear and inconsistent with accepted rules.
If the authors wanted to obtain a non-polar fraction why did they use maceration in a mixture of very polar solvents (methanol and water)?
Besides, it is not very clear what the authors wanted to obtain: an essential oil, an oil or a fraction of volatile compounds?
If they ectracted the base extract with hexane, chloroform, ethyl acetate and butanol, why didn't they test the other fractions (except hexane)?
It is illogical and against the art to apply a non-polar fraction dissolved in methanol to a chromatography column containing silica gel.
Moreover, how was the elution carried out using hexane with increasing amounts of ethyl acetate? After dosing the column with methanol???
The column chromatography technique is used to iosylate compounds or at least fractions.
The authors did not mention what they obtained from the separation of the extract on the column!
It all seems illogical.
Also, what part of the plant was studied? The above-ground parts, the root, the herb? One can only guess from the description that the herb with flowers was used.
The paper also contains a lot of errors
For example, MIC values should be expressed (as concentration) in mg/L or µg/mL. It is a huge mistake to express concentrations in units (µL) (page 7 of manuscript).
Author Response
Response to Reviewer 2
Reviewer Comment: The process of extracting the plant material and the process of obtaining the hexane fraction (so-called by the authors as oil) is unclear and inconsistent with accepted rules.
Author response: Dear Reviewer, thanks for highlighting the mistake. We have corrected the extraction process as per rule.
Reviewer Comment: If the authors wanted to obtain a non-polar fraction why did they use maceration in a mixture of very polar solvents (methanol and water)?.
Author response: Dear Reviewer, our aim was to isolate the pure constituents from the polar and non-polar fractions of the selected plant (phytochemically unexplored). We did solvent-solvent fraction for the isolation of polar compounds. As during column chromatography, we got good amount of oil which showed biological α-glucosidase, antioxidant, anti-inflammatory, and antimicrobial activities. As isolation of pure compounds from the oil is very difficult, therefore, we decided to do the GC-MS analysis to profile the active constituents.
Reviewer Comment: Besides, it is not very clear what the authors wanted to obtain: an essential oil, an oil, or a fraction of volatile compounds?
Author response: Our aim was to isolate pure compounds from the n-hexane fraction of the selected plant after showing promising biological activities. As during column chromatography, we got good amount of oil which showed α-glucosidase, antioxidant, anti-inflammatory, and antimicrobial activities. As isolation of pure compounds from the oil is very difficult, therefore, we decided to do the GC-MS analysis to see the active constituents.
Reviewer Comment: If they extracted the base extract with hexane, chloroform, ethyl acetate, and butanol, why didn't they test the other fractions (except hexane)?
Author response: Dear reviewer, we have done the activity of all the fractions and even isolated compounds from the EtOAc fraction which is already published by our group. In addition, non-polar or volatile compounds are mostly soluble in n-hexane solvent. That’s why we decided to load the n-hexane fraction to get the low polar compounds and luckily obtained oil.
Reviewer Comment: It is illogical and against the art to apply a non-polar fraction dissolved in methanol to a chromatography column containing silica gel.
Author response: Dear reviewer, there was a typographical mistake in the isolation scheme. Now we have corrected the scheme as per isolation rule. We have dissolved non-polar fraction in n-hexane solvent and then loaded over silica gel column chromatography. The oil obtained after passing n-hexane as mobile phase.
Reviewer Comment: Moreover, how was the elution carried out using hexane with increasing amounts of ethyl acetate? After dosing the column with methanol???
Author response: Dear reviewer, the oil was obtained using pure n-hexane solvent as a mobile phase. For detail, kindly see the revised mythology (3.4. Oil extraction).
Reviewer Comment: The column chromatography technique is used to isolate compounds or at least fractions.
Author response: Dear reviewer, column chromatography is mostly used for the isolation of pure compounds, while a mixture of two, three, or more compounds can be obtained. We are working on the n-hexane fraction to look for a new or novel compounds which is under progress and will be published later.
Reviewer Comment: The authors did not mention what they obtained from the separation of the extract on the column! It all seems illogical.
Author response: Dear reviewer, the oil can be obtained from the h-hexane fraction through column chromatography. As I mention above that our main aim was to isolate the compounds from the n-hexane fraction of the plant and, luckily, got oil which showed promising biological activities, that’s why we decided to proceed the GC-MS for the oil to see the active ingredients.
Reviewer Comment: Also, what part of the plant was studied? The above-ground parts, the root, the herb? One can only guess from the description that the herb with flowers was used.
Author response: Dear reviewer, the plant selected for the current project is herbaceous by habit, thus whole plant was used.
Reviewer Comment: The paper also contains a lot of errors, for example, MIC values should be expressed (as concentration) in mg/L or µg/mL. It is a huge mistake to express concentrations in units (µL) (page 7 of the manuscript).
Author response: Dear reviewer, yes you are right. Changes were made accordingly, and the typographic mistake was rectified in all sections and highlighted in the revised manuscript.

Reviewer 3 Report
The research topics are " Fatty acid profile and biomedical therapy of n-Hexane extracted oil of scutellaria edelbergii Rech. F.: in-vitro, in-vivo, and in-silico approach", this manuscript was described n-hexane extracted oil (HEO) from the n-hexane fraction of Scutellaria edelbergii and further analyzed their chemical composition, in vitro antibacterial, antifungal, antioxidant, antidiabetic, and in vivo anti-inflammatory, and analgesic activities, the research result of this manuscript demonstrated the S. edelbergii oil contained valuable bioactive constituents. This research is very interesting and the experimental design very complete. For all these reasons I recommend minor revisions to the manuscript be considered for publication in this scientific journal.
The following remarks should be considered and commented:
- Line 47-51 different font sizes should be corrected
- Line 50 “were44.49” should correct to “were 44.49”
- Figure 1 the number in the GC chromatogram, what is the number mean
- The labels (A), (B), (C), and (D) in Figure 4 are unclear and easy to misunderstand should be corrected
- Line 595, 739 GC-MS or GCMS, please consistence
- Line 628 Equation-2 should correct to Equation 2
- Line 687 what is “for45daysatmaintained”
- The conclusion too long should be shortened
Author Response
Response to Reviewer 3rd
Comments and Suggestions for Authors
Reviewer Comment: The research topics are " Fatty acid profile and biomedical therapy of n-Hexane extracted oil of Scutellaria edelbergii Rech. F.: in-vitro, in-vivo, and in-silico approach", this manuscript was described n-hexane extracted oil (HEO) from the n-hexane fraction of Scutellaria edelbergii and further analyzed their chemical composition, in vitro antibacterial, antifungal, antioxidant, antidiabetic, and in vivo anti-inflammatory, and analgesic activities, the research result of this manuscript demonstrated the S. edelbergii oil contained valuable bioactive constituents. This research is very interesting and the experimental design very complete. For all these reasons I recommend minor revisions to the manuscript be considered for publication in this scientific journal.
Autor response: Thanks, Dear reviewer for appreciating and recommending our work for this well-reputed journal.
The following remarks should be considered and commented:
Reviewer Comment 1. Line 47-51 different font sizes should be corrected
Autor response: Dear reviewer thanks for your valuable comment. Font size of the abstract corrected in the revised manuscript.
Reviewer Comment 2. Line 50 “were44.49” should correct to “were 44.49”.
Autor response: Dear reviewer corrected.
Reviewer Comment 3. Figure 1 the number in the GC chromatogram, what is the number mean.
Autor response: These are the serial number of the compounds. To avoid any confusion, we have deleted the S. No. row from the Figure 1.
Reviewer Comment 4. The labels (A), (B), (C), and (D) in Figure 4 are unclear and easy to misunderstand should be corrected.
Autor response: Dear reviewer, the figure has been defined in the caption of figure insertion in the revised article.
Reviewer Comment 5. Line 595, 739 GC-MS or GCMS, please consistent.
Autor response: It is GC-MS and kept consistency throughout the manuscript as per reviewer suggestion.
Reviewer Comment 6. Line 628 Equation-2 should correct to Equation 2.
Autor response: Worthy reviewer corrected.
Reviewer Comment 7. Line 687 what is “for45daysatmaintained”.
Autor response: Dear reviewer, it was spacing mistake, which is corrected and highlighted in the revised manuscript.
Reviewer Comment 8. The conclusion too long should be shortened.
Autor response: Dear reviewer the conclusion has been reduced as per your suggestion.

Round 2
Reviewer 2 Report
In my opinion the anawers didn not explain my doubts.